

# 1 Unexpectedly high concentrations of atmospheric mercury

# 2 species in Lhasa, the largest city on the Tibetan Plateau

Huiming Lin[1], Yindong Tong[2,3*], Long Chen[4], Chenghao Yu[5], Zhaohan Chu[1], Qianru
Zhang[1], Xiufeng Yin[6], Qianggong Zhang[7,8], Shichang Kang[6,8], Junfeng Liu[1], James
Schauer[9,10], Benjamin de Foy[11], Xuejun Wang[1**]
1. MOE Laboratory of Earth Surface Processes, College of Urban and Environmental Sciences, Peking
University, Beijing 100871, China;
2. School of Science, Tibet University, Lhasa 850000, China
3. School of Environmental Science and Engineering, Tianjin University, Tianjin, 300072, China;
4. School of Geographic Sciences, East China Normal University, Shanghai 200241, China;
5. Key Laboratory of Microbial Technology for Industrial Pollution Control of Zhejiang Province,
College of Environment, Zhejiang University of Technology, Hangzhou 310014, China
6. State Key Laboratory of Cryospheric Science, Northwest Institute of Eco-Environment and Resources,
Chinese Academy of Sciences, Lanzhou 730000, China
7. Key Laboratory of Tibetan Environment Changes and Land Surface Processes, Institute of Tibetan
Plateau Research, Chinese Academy of Sciences, Beijing, 100101, China;
8. CAS Center for Excellence in Tibetan Plateau Earth Sciences, Beijing, 100085, China;
9. Department of Civil and Environmental Engineering, University of Wisconsin-Madison, Madison, WI,
USA;
10. Wisconsin State Laboratory of Hygiene, University of Wisconsin-Madison, WI, USA;
11. Department of Earth and Atmospheric Sciences, Saint Louis University, St. Louis, MO, 63108, USA
**Correspondence:**
*Xuejun Wang (wangxuejun@pku.edu.cn) **Yindong Tong (yindongtong@tju.edu.cn)
**Abstract**
Lhasa City is located in the central Tibetan Plateau and is the most densely populated area. As
the first continuous monitoring of atmospheric mercury (Hg) species in a city on the Tibetan Plateau,
our monitoring in Lhasa showed that the concentrations of gaseous elemental Hg (GEM), gaseous
oxidized Hg (GOM), and particle-bound Hg (PBM) during subsequent of the Indian Summer
Monsoon (S-ISM) period were $2.73 \pm 1.48$ ng m$^{-3}$, $38.4 \pm 62.7$ pg m$^{-3}$, and $59.1 \pm 181.0$ pg m$^{-3}$,
respectively. During the Westerly Circulation (WEC) period, the GEM, GOM and PBM
concentrations were $2.11 \pm 2.09$ ng m$^{-3}$, $35.8 \pm 43.3$ pg m$^{-3}$, and $52.9 \pm 90.1$ pg m$^{-3}$, respectively.
The atmospheric Hg species concentrations were higher than those of previous monitoring on the
Tibetan Plateau and other provincial capitals in China. Typical high-value occurrence processes
were studied to investigate random events with high atmospheric Hg concentrations in Lhasa.





Combustion event nearby or further away may be the main contributor of the high-concentration
events. The lowest GEM concentrations occurred in the afternoon and persistently high
concentrations were observed at night. The changes in GEM concentrations were consistent with
the trends of other pollutant concentrations and contradictory to those of the wind speed. The high
GEM concentrations at night can be attributed to the lower boundary layer height and lower wind
speed. For both GOM and PBM, higher GOM concentrations occurred during the day and PBM
during the night. The results of the principal component analysis indicated that local sources and
wind speed are important factors influencing atmospheric Hg concentrations in Lhasa. The
trajectory simulation showed that the source of the GEM in Lhasa gradually shifted from the south
to the west of Lhasa from the S-ISM to the WEC periods, while both the southern and western
sources were important in the late WEC period. The concentrations and change patterns of Hg
species in Lhasa were significantly different than those at other monitoring sites on the Tibetan
Plateau. Monitoring Hg species in Lhasa shows the possible maximum anthropogenic influences on
the Tibetan Plateau and demonstrates the dramatic effect of wind on changes in urban atmospheric
Hg concentrations.

**1. Introduction**
Mercury (Hg) has received worldwide attention owing to its high toxicity and bioaccumulation.
Atmospheric mercury (Hg) exists in three different forms: atmospheric gaseous elemental Hg
(GEM), gaseous oxidized Hg (GOM), and particle-bound Hg (PBM) (Selin, 2009). They exhibit
different behaviors in the environment owing to their various chemical properties (Selin,
2009;Travnikov et al., 2017;Lindberg and Stratton, 1998;Seigneur et al., 2006). Many established
monitoring networks for atmospheric Hg exist in North America and Europe (Stylo et al., 2016)
including the Atmospheric Mercury Network (AMNet; Gay et al., 2013), the Global Mercury
Observation System (GMOS; Sprovieri et al., 2013;Sprovieri et al., 2016), the Canadian
Atmospheric Mercury Network (CAMNet; Kellerhals et al., 2003), and the Arctic Monitoring
Assessment Programme (AMAP; https://mercury.amap.no/) (Gay et al., 2013;Sprovieri et al.,
2013;Sprovieri et al., 2016;Kellerhals et al., 2003). They have been operating for decades and have
provided a large amount of atmospheric Hg data. Compared to Europe and the United States,



independent research teams have conducted monitoring work in China based on different research
interests (Fu et al., 2012b;Fu et al., 2008;Fu et al., 2016a;Fu et al., 2019;Fu et al., 2016b;Liu et al.,
2011;Feng and Fu, 2016;Feng et al., 2013;Wang et al., 2015;Hu et al., 2014;Ci et al., 2011;Duan et
al., 2017;Liu et al., 2002;Yin et al., 2018;Yin et al., 2020b;Lin et al., 2022;Lin et al., 2019). Most
monitoring stations are set up only in developed regions, such as eastern and central China, owing
to operational difficulties in remote areas. Few studies on atmospheric Hg in western China have
been reported; thus, little is known about the overall level of atmospheric Hg in western China. To
better employ the Minamata Convention and verify the effect of the implementation of the
Convention, monitoring atmospheric Hg concentrations around the globe is significant and can aid
in identifying the global Hg transport pattern.

75        The Tibetan Plateau is in the mid-latitudes of the Northern Hemisphere (in central Asia) and is

an important area for studying the global Hg circulation. Owing to the high altitude and rough living
conditions, there is little Hg research on the Tibetan Plateau. This area is less developed and there
are few industrial activities; therefore, it is generally considered a clean region and can be treated
as a background condition. However, there are large tourist cities in this area, such as Lhasa, where
the number of tourists reached 40,121,522 in 2019 (Tibet Bureau of Statistics, 2020). Local cement
production in Tibet reached 10.81 million tons in 2019 (Tibet Bureau of Statistics, 2020).
Meanwhile, although the high altitude makes the Tibetan region a natural barrier between inland
China and the Indian subcontinent (Qiu, 2008;Yao et al., 2012;Pant et al., 2018), the Tibetan Plateau
is potentially influenced by the Indian summer monsoon (ISM) and the Westerly circulation (WEC).
Trans-boundary inputs of atmospheric pollutants to the Tibetan Plateau have been demonstrated in
pollutant studies such as with persistent organic pollutants and black carbon (Yang et al., 2018;Li et
al., 2016a;Zhang et al., 2015b;Pokhrel et al., 2016;Wang et al., 2018;Zhang et al., 2015a;Feng et al.,
2019;Zhu et al., 2019). Our previous study on atmospheric Hg in the Qomolangma region (QNNP)
also suggested that atmospheric Hg from India can be transported and affect atmospheric Hg
concentrations on the Tibetan Plateau as a result of the Indian monsoon (Lin et al., 2019). Hence, it
remains unclear whether the Tibetan Plateau can be treated as a background area for studying
atmospheric Hg, and further monitoring data are required. Monitoring in the largest cities on the
Tibetan Plateau will provide important information and corroboration to address this query.



In previous study, Yin et al. (2018) reported GEM concentration data for the Namco region on
the Tibetan Plateau from 2012-2014 and found that the GEM concentration at Namco was 1.33 ±
0.24 ng m$^{-3}$, which is lower than the mean GEM concentration in the Northern Hemisphere
(Lindberg et al., 2007;Slemr et al., 2015;Venter et al., 2015;Sprovieri et al., 2016;Lan et al., 2012).
Our previous study at QNNP (Lin et al., 2019) showed that the atmospheric Hg concentrations were
1.42 ± 0.37 ng m$^{-3}$, 21.4 ± 13.4 pg m$^{-3}$, and 25.6 ± 19.1 pg m$^{-3}$ for GEM, GOM, and PBM,
respectively, close similar to the average GEM concentrations in the Northern Hemisphere. The
concentrations of atmospheric Hg species in Nyingchi, in the southeast Tibetan Plateau, were very
low (1.01±0.27 ng m$^{-3}$, 12.8±13.3 pg m$^{-3}$, and 9.3±5.9 pg m$^{-3}$ for GEM, GOM, and PBM,
respectively), which may be affected by heavy wet deposition and the large amounts of vegetation
in the Yarlung Zangbu/Brahmaputra Grand Canyon (Lin et al., 2022). However, Namco, QNNP
and Nyingchi are remote areas on the Plateau, with few populations and industries. During previous
studies in Lhasa, the largest city on the Plateau, only dry and wet depositions of atmospheric Hg
were analyzed. Monitoring of atmospheric Hg particulate matter (Huang et al., 2016a) indicated that
Lhasa has mean particulate Hg levels as high as 224 pg m$^{-3}$ (ranging from 61.2 to 831 pg m$^{-3}$),
which is much higher than expected. Huang et al. (2013) measured the wet deposition of
atmospheric Hg in Lhasa in 2010 and showed that the wet deposition of total Hg and particulate Hg
was higher during the non-monsoon period than during the monsoon period. The active Hg was
higher during the monsoon than during the non-monsoon period, and they concluded that the wet
deposition of Hg originated mainly from local sources. This indicates that atmospheric Hg
concentrations in Lhasa may be elevated and further detailed monitoring is needed.
In this study, we conducted a high-time-precision atmospheric Hg species monitoring system
in Lhasa. We performed cont inuous measurements of GEM, GOM, and PBM concentrations from
subsequent of the Indian Summer Monsoon (S-ISM) period to the WEC period from 2016 to 2017.
Based on literature research, this is the first continuous monitoring of atmospheric Hg species in a
city on the Tibetan Plateau, and the influence of human activities, meteorological factors, and long-
range transportation of pollutants on the diurnal variation of atmospheric Hg in Lhasa is discussed.
We combined monitoring with other pollutant concentrations to explore the main factors influencing
the local atmospheric Hg concentrations. To determine the detailed source profile of atmospheric





Hg, we combined real-time Hg monitoring data with backward trajectory and cluster analyses. This
study can help understand atmospheric Hg characteristics in the city of the Plateau and provide
scientific support for managerial decision-making.
**2.  Material and methods**
**2.1  Atmospheric Hg monitoring sites**
The monitoring site for atmospheric Hg species in Lhasa was set up on the top floor of the
Lhasa station office building in the Institute of Tibetan Plateau Research, Chinese Academy of
Sciences, in western Lhasa City (29.64°N, 91.03°E, 3650 m above sea level; Figure 1, Figure S1).
Lhasa is located in the central region and is the largest city on the Tibetan Plateau, covering an area
of approximately 60 km$^2$. The Lhasa population in 2019 was 720,700, accounting for approximately
20.6% of the total population of the Tibet Autonomous Region (Tibet Bureau of Statistics, 2020).
The entire city is in a flat river valley surrounded by mountains up to 5,500 m above sea level.
During the ISM period (from May to September), the low pressure on the Tibetan Plateau attracts
the summer monsoon from the Indian Ocean to the Plateau, exhibiting a wetter monsoon season
(Qiu, 2008). During the non-monsoon season (from October to April), the large-scale atmospheric
circulation on the Tibetan Plateau is mainly under the control of westerly winds, which largely come
from the inland areas of Central Asia, presenting a drier season in Lhasa during this time (Huang et
al., 2010;Guo et al., 2015). According to previous studies, the air quality in Lhasa may be influenced
by local emissions from anthropogenic activities (e.g., power plants, cement facilities, vehicular
traffic, and religious activities) (Li et al., 2008;Cong et al., 2011;Huang et al., 2010;Guo et al.,
2015;Luo et al., 2016;Li et al., 2016b) and long-range transboundary atmospheric transport (Huang
et al., 2016a;Huang et al., 2016b).

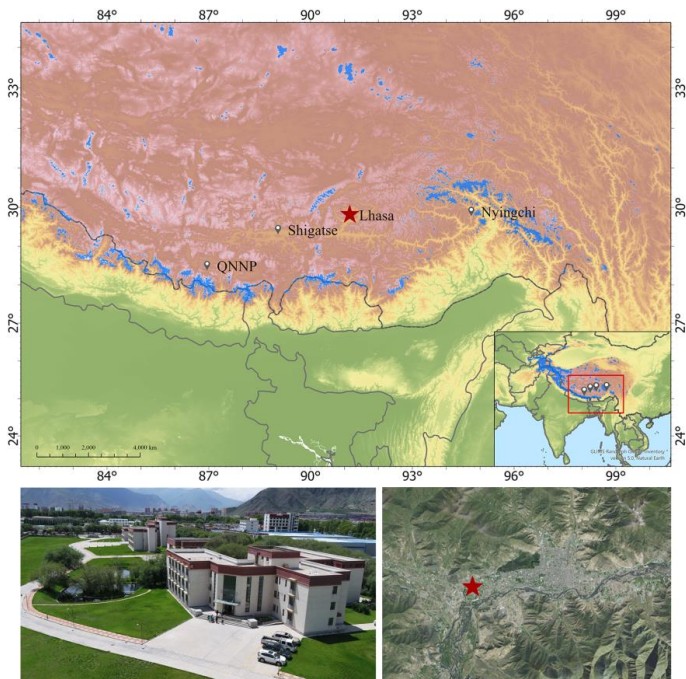


**Figure 1 Location of the Lhasa station in the Institute of Tibetan Plateau Research,**

**Chinese Academy of Sciences (red star).**


### 2.2 GEM, GOM, and PBM active monitoring

Monitoring in Lhasa was performed using Tekran models 2537 B, 1130, and 1135 (Tekran Inc.,
Toronto, Canada) for real-time continuous measurements of GEM, GOM, and PBM concentrations.
Model 2357b is the main analytical module used to analyze Hg concentrations employing the cold
atomic fluorescence technique. Model 1130 is divided into pump and lysimeter modules, which is
mainly used for the collection and resolution of GOM. Model 1135 is the particle collection module
that is mainly used to collect and analyze atmospheric Hg in the particulate state. During the actual
monitoring, considering the low air pressure on the Tibetan Plateau, we reduced the airflow of the
pump module to 7.5 L/min (Swartzendruber et al., 2009;Zhang et al., 2015a;Zhang et al., 2016;Lin
et al., 2019;Lin et al., 2022) and the airflow in model 2537B was reduced to 1 L/min to ensure that
atmospheric Hg monitoring could be continuously performed. All monitoring data were converted
to concentrations under standard atmospheric pressure. The Tekran 2537B analyzer was





automatically calibrated every 23 hours using the instrument's internal Hg permeation source and
was calibrated before and after monitoring using an external Hg source. The same instrument setup
was used for QNNP and Nyingchi (Lin et al., 2019;Lin et al., 2022). The Tekran ambient Hg
analyzer has been described in detail in previous studies (Landis et al., 2002; Rutter et al. 2008; de
Foy et al. 2016).
**2.3  Meteorological data and other pollutant data**
During the monitoring period, the Vantage Pro2 weather station (Davis Instruments, USA)
recorded local temperature (accuracy of 0.1°C), relative humidity (accuracy of 1%), wind speed
(accuracy of 0.1 m s$^{-1}$), wind direction (accuracy of 1°), barometric pressure (accuracy of 0.1 hPa),
solar radiation (accuracy of 1 W m$^{-2}$), and UV index (accuracy 0.1 MEDs). Hourly measurements
of $PM_{2.5}$, $PM_{10}$, $SO_2$, $NO_2$, $O_3$, and CO concentrations and the air quality index (AQI) were obtained
from monitoring stations hosted by the Ministry of Ecology and Environment of China and
published by the China Environmental Monitoring Center. The station was set up 10 km from the
atmospheric Hg monitoring station.
**2.4  Backward trajectory simulation and potential source analysis**
To better understand the transport paths of atmospheric GEM, the Hybrid Single-Particle
Lagrangian Integrated Trajectory (HYSPLIT) model was used to calculate backward trajectories
(Stein et al., 2015;Chai et al., 2017;Chai et al., 2016;Hurst and Davis, 2017;Lin et al., 2019). The
HYSPLIT model (https://www.arl.noaa.gov/hysplit/hysplit/) is a hybrid approach that combines
Lagrangian and Eulerian methods, which was developed by the National Oceanic and Atmospheric
Administration (NOAA) as a tool to explain the transport, dispersion, and deposition of particles in
the atmosphere. The backward trajectory simulation used Global Data Assimilation System (GDAS)
data with 1°x1° latitude and longitude horizontal spatial resolution and 23 vertical heights every 6
hours in this study. We examined the effect of arrival height on the trajectories using different arrival
heights (50 m, 100 m, 400 m, and 1,000 m) in December 2016 (Figure S2). The results showed that
the calculated trajectories of the air masses were almost the same when the arrival height was below
400 m. The trajectory arrival altitude was then set to 100 m a.g.l in this study. The trajectories were
computed every 6 hours with an inverse time of 120 hours. The trajectories could cover China,
Nepal, India, Pakistan, and most of West Asia. Here, we combined the backward trajectory with





real-time Hg monitoring concentrations to represent the trajectories of GEM concentrations. Cluster
analysis was performed after the trajectory calculation. Cluster analysis can indicate the main
trajectory of incoming pathways and the GEM concentration indicated by the incoming trajectories.
**2.5 Principal component analysis**
Principal component analysis (PCA) is a data reduction method that allows a number of
measured variables to be catergorized into several factors that represent the behavior of the entire
dataset (Jackson, 2005). In many previous Hg studies, PCA has been used to analyze the
relationships between Hg and multiple pollutants and meteorological variables (Brooks et al.,
2010;Cheng et al., 2012;Liu et al., 2007;Zhou et al., 2019;Lin et al., 2022). Prior to running the
PCA, all variables were normalized by the standard deviation. To check whether PCA was the
appropriate method for the dataset used in this study, Kaiser-Meyer-Olkin's measure of sampling
adequacy (MSA>0.5) and Bartlett's Test of sphericity (P<0.05) tests were performed during data
analysis. Total variance and rotated scree plots were used to determine the number of factors during
the PCA analysis, and components with variance ≥1.0 were retained. Variables with high factor
loadings (generally >0.5) were identified as potential sources of Hg in this study.
**3.    Results and Discussion**
**3.1   Atmospheric Hg Monitoring in Lhasa**
Atmospheric Hg monitoring in Lhasa comprised the subsequent Indian Summer Monsoon (S-
ISM) and Westerly circulation (WEC) periods. Figure 2 shows the variation in atmospheric Hg
concentrations at the station during the monitoring period. During the whole monitoring period, the
mean concentrations of GEM, GOM, and PBM at the station were 2.26±1.97 ng m$^{-3}$, 36.4±48.9 pg
m$^{-3}$, and 54.5±119.5 pg m$^{-3}$ (mean concentration ± standard deviation), respectively. During the S-
ISM period, the concentrations of GEM, GOM, and PBM were 2.73±1.48 ng m$^{-3}$, 38.4±62.7 pg m$^{-}$
$^{3}$, and 59.1±181.0 pg m$^{-3}$, respectively, while during the WEC period, the concentrations of GEM,
GOM, and PBM were 2.11±2.09 ng m$^{-3}$, 35.8±43.3 pg m$^{-3}$, and 52.9±90.1 pg m$^{-3}$, respectively. The
GEM concentrations during the S-ISM period were significantly higher than those during the WEC
period (p < 0.01), while the mean concentrations of GOM and PBM in the S-ISM period were
slightly higher than those in the WEC period. Overall, GEM concentrations showed a decreasing
trend throughout the monitoring period, with the average weekly concentration decreasing from



3.21 ng m$^{-3}$ at the beginning of the monitoring period to 1.60 ng m$^{-3}$ at the end of the monitoring
period. Unexpectedly high concentrations were found at irregular intervals for all Hg species. The
occurrence time of these high concentrations was random, and high GEM concentrations did not
always occur at the same time as high GOM or PBM concentrations, indicating the complexity of
the Hg sources of the species. For GOM and PBM, relatively comparable trends between them may
be related to similar sources, transport, and transformation reactions in the atmosphere.

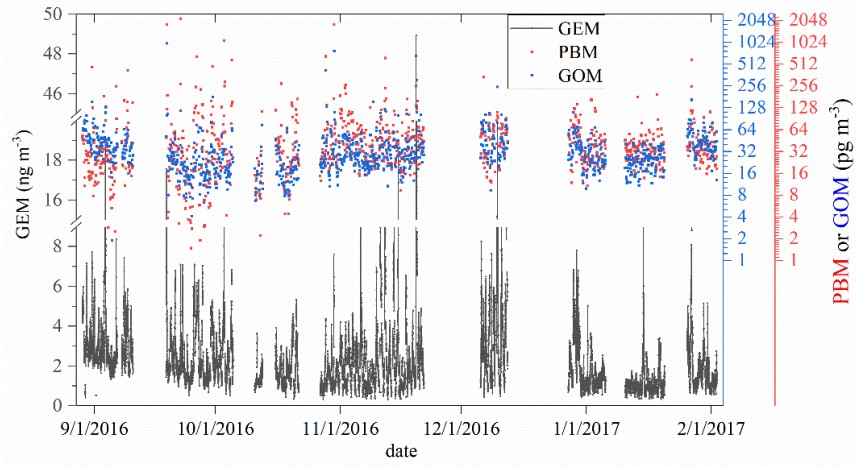


**Figure 2 Time series of GEM, GOM, and PBM concentrations over the sampling period. The**
**GEM concentration resolution was 5 min and the GOM and PBM resolutions was 2 h.**

229        In early September, GOM concentrations were generally higher than PBM concentrations. In

the subsequent period, PBM concentrations were always higher than GOM concentrations, which
may indicate that the sources and composition of pollutants at this time were not consistent with
those in the latter period. The GOM and PBM concentrations were higher in November and
December. Since GOM and PBM are mainly from local emissions, the changes in their
concentrations may indicate that there are more local sources at this period. As Lhasa enters the
heating season in November-December, and there are more local religious activities at this time,
there may be more local combustion sources. Table 1 lists the concentrations of Hg and other
pollutants during the monitoring period. PM$_{2.5}$ concentration increased significantly in the WEC1
period, indicating the presence of more particulate matter during this period. This could explain the





elevated concentrations of GOM and PBM from November-December.

Table 1 Statistics metrics of species Hg, meteorological factors, and other pollutants

| Period | Stat | GEM (ng m⁻³) | PBM (pg m⁻³) | GOM (pg m⁻³) | Temp (°C) | Hum. (%) | Wind speed (m s⁻¹) | Solar radiation (W m⁻²) | CO (mg m⁻³) | NO₂ (µg m⁻³) | O₃ (µg m⁻³) | PM₂.₅ (µg m⁻³) | SO₂ (µg m⁻³) |
|---|---|---|---|---|---|---|---|---|---|---|---|---|---|
|  | Mean | 2.73 | 59.08 | 38.39 | 14.42 | 61.45 | 1.70 | 212.60 | 0.39 | 26.44 | 72.18 | 16.26 | 4.19 |
|  | SD | 1.48 | 181.38 | 62.85 | 3.70 | 16.96 | 1.41 | 313.90 | 0.18 | 19.36 | 28.68 | 13.12 | 2.08 |
| S-ISM | Median | 2.36 | 24.50 | 30.30 | 13.90 | 62.00 | 1.30 | 7.00 | 0.30 | 22.00 | 77.00 | 13.00 | 4.00 |
|  | Min | 0.40 | -11.44 | 1.89 | 5.90 | 14.00 | 0.00 | 0.00 | 0.10 | 1.00 | 1.00 | 1.00 | 1.00 |
|  | Max | 18.87 | 2165.70 | 988.50 | 24.70 | 93.00 | 13.00 | 1290.00 | 1.70 | 110.00 | 133.00 | 81.00 | 21.00 |
|  | Mean | 2.39 | 57.55 | 37.46 | 6.90 | 26.01 | 1.28 | 169.56 | 0.64 | 45.24 | 45.39 | 47.97 | 5.51 |
|  | SD | 2.35 | 104.15 | 50.98 | 5.75 | 15.08 | 1.39 | 246.66 | 0.51 | 31.01 | 34.44 | 46.19 | 5.35 |
| WEC1 | Median | 1.79 | 38.80 | 27.83 | 6.90 | 24.00 | 0.90 | 0.00 | 0.50 | 41.00 | 43.00 | 35.00 | 5.00 |
|  | Min | 0.31 | 0.00 | 1.89 | -6.20 | 1.00 | 0.00 | 0.00 | 0.10 | 2.00 | 1.00 | 1.00 | 1.00 |
|  | Max | 48.93 | 1797.30 | 774.86 | 21.60 | 94.00 | 9.80 | 973.00 | 4.40 | 152.00 | 133.00 | 458.00 | 145.00 |
|  | Mean | 1.47 | 42.82 | 32.05 | 0.22 | 25.59 | 1.97 | 123.61 | 0.56 | 26.78 | 65.22 | 23.73 | 4.49 |
|  | SD | 1.12 | 45.39 | 17.14 | 3.98 | 14.67 | 1.85 | 186.98 | 0.30 | 22.61 | 26.30 | 18.77 | 2.61 |
| WEC2 | Median | 1.17 | 33.77 | 27.50 | 0.20 | 24.00 | 1.30 | 0.00 | 0.50 | 19.00 | 71.00 | 19.00 | 4.00 |
|  | Min | 0.33 | 0.00 | 9.91 | -8.80 | 2.00 | 0.00 | 0.00 | 0.20 | 2.00 | 1.00 | 1.00 | 1.00 |
|  | Max | 20.86 | 589.43 | 165.06 | 10.20 | 68.00 | 11.20 | 662.00 | 2.60 | 99.00 | 108.00 | 118.00 | 32.00 |


Table 2 shows the distributions of atmospheric Hg concentrations in some provincial capitals
in China and nearby monitoring stations from literature. In general, the GEM concentration in Lhasa
is low among the provincial capitals in China. The GEM concentration in other provincial capitals
of China was approximately 3-10 ng m⁻³. Guiyang, Chongqing, and Lanzhou were the nearest





provincial capitals to Lhasa, with GEM concentrations reported in the literature, all located in
western China. Guiyang had a very high GEM concentration due to the presence of local Hg mines
(Liu et al., 2011;Yang et al., 2009). The GEM concentration in Chongqing was approximately three
times higher than that of Lhasa. The higher GEM concentration in Chongqing was likely due to its
proximity to the Hg-contaminated area and large population (Yang et al., 2009). Compared to
Lanzhou, another high-altitude city, the GEM concentration in Lhasa was approximately half that
of Lanzhou, which may be owing to the overall cleaner environment with fewer local pollution
sources in Lhasa (Yin et al., 2020a). The GOM and PBM concentrations in other provincial capitals
were relatively less monitored. However, GOM concentrations in Lhasa were significantly higher
than those in cities with GOM monitoring, such as mega-cities like Beijing and Shanghai, and even
higher than those in Guiyang, where Hg mines are located. GOM concentrations in provincial
capitals nationwide were mainly concentrated between 3-20 pg m$^{-3}$, whereas the GOM
concentrations in Lhasa were approximately 2-10 times higher than the average concentration in
provincial capitals. The high GOM concentration in Lhasa is likely due to its high altitude. Lhasa is
exposed to much higher solar radiation and has more ice surfaces than inland areas, which may have
contributed to the oxidation of GEM or the re-release of GOM deposited in snow and ice (Steffen
et al., 2008;Dommergue et al., 2003;Song et al., 2018). In contrast, the PBM concentration in Lhasa
was at a lower level, only somewhat higher than that in Hefei. The monitoring period in Lhasa was
mainly in winter when there were more particulate matter emissions than in summer owing to
heating combustion. The PM$_{2.5}$ concentration in Lhasa was low throughout the monitoring period
(Table 1), which indicated that the local particulate matter emissions were low; this may be the main
reason for the low PBM concentration.



Table 2 Comparison of atmospheric Hg concentrations at some provincial capitals in China and some nearby monitoring stations

| location | altitude | type | region | Monitoring period | GEM (ng m⁻³) | GOM (pg m⁻³) | PBM (pg m⁻³) | GEM diurnal variation (local time) peak | valley | variation | reference |
|---|---|---|---|---|---|---|---|---|---|---|---|
| **Lhasa** | **3600** | **City** | **Southwest** | **8/2016-2/2017** | **2.26±1.97** | **36.4±48.9** | **54.5±119.5** | | | | **This study** |
| Beijing | 40 | City | North china | 12/2008-11/2009 | 3.22±1.74 | 10.1±18.8 | 98.2±112.7 | | | | (Zhang et al., 2013) |
| Hefei | 30 | City | East china | 7/2013-6/2014 | 4.07±1.91 | 3.67±5.11 | 30.0±100.3 | | | | (Hong et al., 2016) |
| Shanghai | 4 | City | East china | 2014 | 4.19±9.13 | 21±100 | 197±877 | | | | (Duan et al., 2017) |
| Lanzhou | 1525 | City | Northwest | 10/2016-10/2017 | 4.48±2.32 | - | - | | | | (Yin et al., 2020a) |
| Jinan | 148 | City | East china | 10/2015 -7/2016 | 4.91±3.66 | - | 451.9±433.4 | | | | (Li et al., 2017) |
| Chongqing | 300 | City | Southwest | 2006-2007 | 6.74±0.37 | - | - | | | | (Yang et al., 2009) |
| Nanjing | 25 | City | East china | 2011 | 7.9±7.0 | - | - | | | | (Zhu et al., 2012) |
| Guiyang | 1150 | City | Southwest | 8/2009-12/2009 | 9.72±10.2 | 35.7±43.9 | 368±676 | | | | (Liu et al., 2011) |
| Ev-K2, Nepal | 5050 | Remote | | 11/2011-4/2012 | 1.2±0.2 | - | - | 18/1.30 | 6/1.10 | 0.20 | (Gratz et al., 2013) |
| Nam Co, China | 5300 | Remote | | 11/2014-3/2015 | 1.33±0.24 | - | - | - | - | - | (Yin et al., 2018) |
| Waliguan, China | 3816 | Remote | | 9/2007-9/2008 | 1.98±0.98 | 7.4±4.8 | 19.4±18.1 | 6/2.30 | 14/1.94 | 0.36 | (Fu et al., 2012a) |
| Shangri-La, China | 3580 | Remote | | 11/2009-11/2010 | 2.51±0.73 | 8.22±7.9 | 38.32±31.26 | 17/2.48 | 6/1.71 | 0.77 | (Zhang et al., 2015a) |
| Gongga, China | 1640 | Remote | | 5/2005-6/2006 | 3.98 | - | - | 11/4.45 | 2/3.55 | 0.90 | (Fu et al., 2008) |
| QNNP, China | 4267 | Remote | | 4/2016-8/2016 | 1.42±0.37 | 21.4±13.4 | 25.6±19.1 | 6/2.04 | 13/1.11 | 0.93 | (Lin et al., 2019) |
| Nyingchi, China | 3263 | Remote | | 3/2019-9/2019 | 1.01±0.27 | 12.8±13.3 | 9.3±5.9 | 20/1.07 | 6/0.96 | 0.11 | |





Compared to nearby monitoring stations (Table 2), Hg species concentrations in Lhasa were
high. Namco station is the nearest station; its altitude is 4,730 m and the distance between the two
stations is approximately 120 km. The GEM concentration in Namco was 1.33±0.24 ng m$^{-3}$, which
was only 59% of Lhasa. This is likely because the Namco region is sparsely populated with minimal
local pollution and is far from major Hg pollution sources (Yin et al., 2018). Compared to the QNNP
(Lin et al., 2019) on Mt. Everest, which is approximately 500 km apart, the GEM, GOM, and PBM
concentrations in Lhasa were approximately 1.6, 1.7, and 2.1 times higher, respectively. Our
previous studies demonstrated that the QNNP is influenced by transported air masses from the
Indian subcontinent, indicating that the concentration in Lhasa is high in the Tibetan Plateau.
Compared to another typical highland site, Nyingchi, Lhasa had much high levels of atmospheric
Hg species, which may be related to the vegetation uptake effects and strong wet deposition in
Nyingchi (Lin et al., 2022). Among the surrounding stations, only Mt. Gongga and Shangri-La
stations had higher GEM concentrations than Lhasa. The GEM concentrations reported at Mt.
Gongga station ranged from May 2005 to July 2006. Considering that the smelting activities near
this site were crude at that time and there were almost no air pollution control measures, the high
local GEM concentrations may be strongly affected by local smelting activities and fuel combustion
(Fu et al., 2008). In contrast, GEM concentrations in the Shangri-La region were mainly controlled
by the monsoon, and Zhang et al. (2015a) suggested that all local GEM above 2.5 ng m$^{-3}$ are
associated with the transport of dry air carrying domestic and foreign regional anthropogenic
emissions. Comparing these sites only for the monsoon period, the GEM concentration in Lhasa
was higher than that in the Shangri-La region. As for GOM and PBM, the concentrations at the
Lhasa station were much higher than those in the surrounding areas. The average GOM
concentration in the surrounding areas was approximately 10 pg m$^{-3}$, which was only 27% of that
in Lhasa, and the average PBM concentration in the surrounding areas was approximately 28 pg m$^{-}$
$^{3}$, which was only 54% of that in Lhasa. Considering that GOM and PBM are mainly from local or
surrounding sources or atmospheric transport (Lindberg and Stratton, 1998;Seigneur et al.,
2006;Lynam et al., 2014), high GOM and PBM concentrations may indicate additional local sources
of Hg in Lhasa.
**3.2  Unexpected high concentration events in Lhasa**





To investigate the reasons for the random high atmospheric Hg concentration events in Lhasa,
typical high-value occurrence processes (defined as GEM concentrations above 10 ng m$^{-3}$ monitored
more than five times on a single day) were selected for analysis in the S-ISM and WEC periods,
respectively. A total of seven high GEM concentration events were identified, of which, September
3, November 10, November 19, and December 9, 2016, were selected for analysis; September 18,
October 3, 2016, and January 27, 2017 were omitted due to lack of meteorological data or Hg
concentration data for the proximity date.
During the S-ISM period (Figure 2), there was a clear peak in Hg concentration on September
3, while the GEM, GOM, and PBM concentrations were approximately 1.6, 1.5, and 2.3 times the
average daily value, respectively. Comparing the two days before and after the high-concentration
event (Table 3, Figure 3. a), the concentrations of the three pollutants NO$_2$/PM$_{2.5}$/SO$_2$, were higher
on September 3. High GEM concentrations were accompanied by winds of 2.12 m/s from the
southwest, with NO$_2$ and SO$_2$ concentrations higher than usual as well as increased PBM
concentrations. NO$_2$ and SO$_2$ are typical combustion source pollutants, and the presence of PM$_{2.5}$,
and PBM may indicate more combustion sources in the day. Thus, it can be inferred that the elevated
Hg concentration event on September 3 may have originated mainly from a combustion event
nearby or further away.
During the WEC period, significantly high values were observed on November 10, November
19, and December 9. On November 10 (Figure 3.b), the increase in atmospheric Hg species
concentrations    was    accompanied    by    significant    increases    in    CO/NO$_2$/PM$_{10}$/PM$_{2.5}$/SO$_2$
concentrations. Two similar peaks in atmospheric Hg species concentrations were also observed
around November 10, with relatively lower peak concentrations. During these three events, the
concentrations of other pollutants were higher than usual, whereas wind speeds were extremely low
during the event periods. In addition, extremely high PBM concentrations (297.7±189.3 pg m$^{-3}$,
maximum 621.2 pg m$^{-3}$) were observed at midnight on November 12 which, considering the
extremely low wind speed and the presence of a low height nocturnal boundary layer, may indicate
that the high concentrations originated from local sources.



Table 3 Comparation of the pollutant concentrations with the two days before and after the high Hg concentration events

| Item | | 9.1-2 | 9.3 | 9.4-5 | 11.8-9 | 11.10 | 11.11-12 | 11.17-18 | 11.19 | 11.20-21 | 12.7-8 | 12.9 | 12.10-11 |
|---|---|---|---|---|---|---|---|---|---|---|---|---|---|
| GEM | (ng m$^{-3}$) | 2.72 | 4.63 | 3.03 | 1.47 | 3.16 | 2.96 | 2.15 | 6.87 | 2.66 | 2.76 | 5.43 | 3.24 |
| PBM | (pg m$^{-3}$) | 25.32 | 64.17 | 31.52 | 36.31 | 47.21 | 94.80 | 53.41 | 74.78 | 55.40 | 38.71 | 66.74 | 49.05 |
| GOM | (pg m$^{-3}$) | 40.95 | 63.84 | 42.55 | 32.76 | 39.48 | 42.03 | 35.29 | 119.39 | 44.86 | 43.84 | 69.65 | 43.75 |
| Temp. | (°C) | 16.61 | 15.86 | 15.32 | 7.06 | 7.76 | 8.80 | 4.20 | 4.49 | 4.94 | 3.13 | 2.88 | 2.77 |
| Hum. | (%) | 58.47 | 67.89 | 68.71 | 19.96 | 22.02 | 21.04 | 21.82 | 23.35 | 25.67 | 23.50 | 26.86 | 26.09 |
| Wind | (m s$^{-1}$) | 2.37 | 1.46 | 1.49 | 1.47 | 0.93 | 1.17 | 1.05 | 0.71 | 0.75 | 0.88 | 0.89 | 0.67 |
| Barometer | (hPa) | 651.62 | 651.08 | 652.39 | 654.90 | 654.33 | 652.31 | 655.11 | 652.18 | 653.57 | 657.99 | 655.66 | 653.08 |
| rainfall | (mm) | 0.01 | 0.17 | 0.36 | 0.00 | 0.00 | 0.00 | 0.00 | 0.00 | 0.00 | 0.00 | 0.00 | 0.00 |
| Solar Rad. | (W m$^{-2}$) | 241.20 | 201.15 | 214.39 | 189.02 | 194.45 | 182.30 | 166.14 | 161.59 | 180.02 | 141.61 | 139.69 | 136.64 |
| CO | (mg m$^{-3}$) | 0.34 | 0.36 | 0.32 | 0.39 | 0.76 | 0.63 | 0.64 | 0.71 | 0.72 | 0.67 | 0.80 | 0.74 |
| NO$_2$ | (μg m$^{-3}$) | 20.44 | 26.00 | 21.77 | 41.38 | 57.64 | 66.48 | 54.58 | 53.58 | 55.27 | 48.87 | 55.33 | 56.23 |
| O$_3$ | (μg m$^{-3}$) | 85.73 | 77.42 | 72.56 | 59.27 | 57.43 | 49.46 | 34.73 | 34.79 | 29.51 | 31.37 | 28.38 | 26.81 |
| PM$_{10}$ | (μg m$^{-3}$) | 33.90 | 32.78 | 26.95 | 80.71 | 130.07 | 155.44 | 107.98 | 119.79 | 146.29 | 125.82 | 137.74 | 143.13 |
| PM$_{2.5}$ | (μg m$^{-3}$) | 11.98 | 17.10 | 10.98 | 31.20 | 75.07 | 64.85 | 51.21 | 50.13 | 53.83 | 62.80 | 65.46 | 62.71 |
| SO$_2$ | (μg m$^{-3}$) | 4.23 | 5.89 | 4.91 | 3.83 | 6.07 | 5.71 | 6.50 | 8.92 | 9.44 | 6.00 | 6.88 | 6.69 |


On November 19 (Figure 3.c), there was a significant increase in all atmospheric Hg species concentrations. During the elevated period, the average GEM, GOM, and PBM concentrations were 17.94±10.54 ng m$^{-3}$, 302.5±218.5 pg m$^{-3}$, and 162.4±54.0 pg m$^{-3}$, respectively, which were 8.3, 5.7,





and 4.6 times higher than the average concentrations on November 17-18. However, only a slight
increase in the peak CO/ PM$_{2.5}$/SO$_2$ concentration was observed during the event period. Therefore,
the increase in atmospheric Hg species concentrations may indicate that this event was unique, with
certain sources of high Hg concentrations. The low wind speed and low nocturnal boundary layer
height both contributed to the accumulation of atmospheric Hg species, which also indicated that
the elevated Hg was from local sources. Similar to the event on November 19, sharp increases in
atmospheric Hg species concentrations were observed only on December 9 (Figure 3.d). The
atmospheric Hg species concentration increase events during the WEC period occurred at night and,
given that it coincided with the winter heating period, local combustion sources may be the unique
sources that may contribute to the Hg concentration increase.

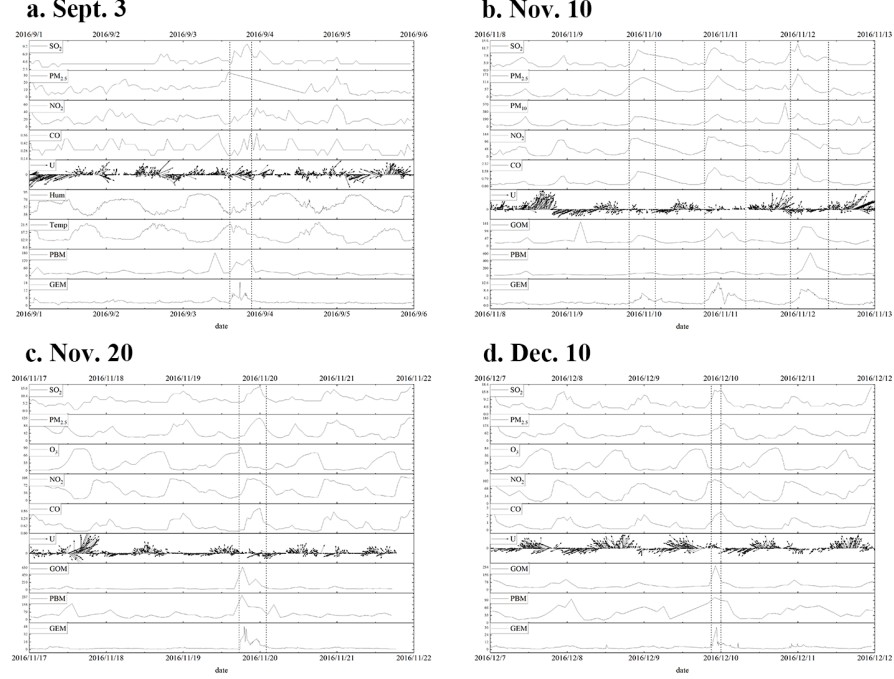


**Figure 3 Comparison of the pollutant concentrations with the two days before and after the**
**high Hg concentration events**
**3.3 Diurnal variation of atmospheric Hg in Lhasa**
Figure 4.a shows the diurnal variation of atmospheric Hg in Lhasa during the monitoring period.
The mean GEM concentration during this period was 2.26±1.97 ng m$^{-3}$, and the maximum





difference of hourly average GEM diurnal variation was 1.89 ng m$^{-3}$. GEM concentration was low
during the day; the lowest concentration of the day appeared from 14:00-18:00 (UTC+8). The GEM
concentration kept increasing and reached the peak at midnight. During the night, the GEM
concentration maintained in high value with little dissipation. Subsequently, the GEM concentration
decreased rapidly with the sunrise. Overall, the diurnal variation of GEM concentration was similar
to CO, NO$_2$, and relative humidity, and opposite to O$_3$ and wind speed. The high GEM
concentrations at night can be attributed to the lower boundary layer height at night (average at 131
m a.g.l. during the observation period, between 19:00 and 07:00, UTC+8) and the lower wind speed
(Figure 4). Also, the average night temperature in Lhasa during the monitoring period was 5.8℃,
and the residents heating combustion may bring about the release of GEM. After sunrise, the
boundary layer height increased rapidly and the diffusion quickly occurred in the lower atmosphere.
During the increase of wind speed, airflows were carried from other regions with few populations,
which may lead to a decrease in GEM concentrations. For both GOM and PBM, the higher GOM
concentrations occurred during the day and higher PBM concentration occurred during the night.
No obvious increase occurred for GOM and PBM concentrations at the same time, which may
indicate that there is no local common external source, and the variation of GOM and PBM
concentrations may come from the gas-particle redistribution process of atmospheric Hg(II).

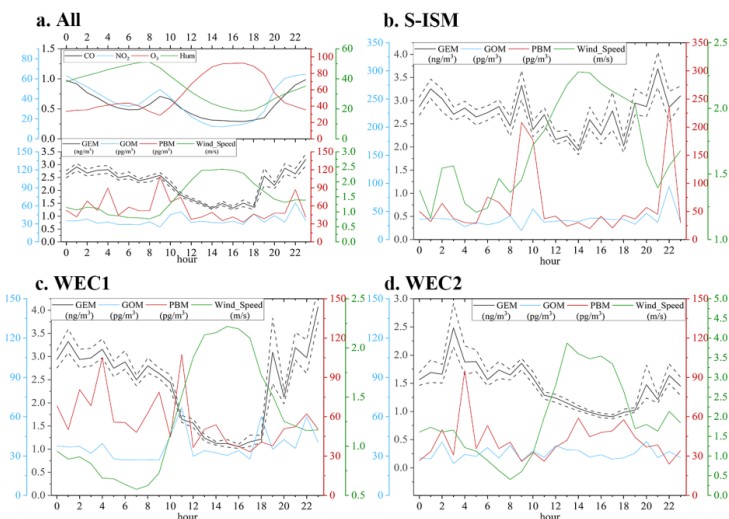


**Figure 4 Diurnal variations of Hg species, concentrations of other pollutants, and**
**meteorological information from S-ISM to WEC periods. The short horizontal line**





**represents the concentration error range for each time period.**
In particular, during the S-ISM period (Figure 4.b), the diurnal variation in GEM
concentrations fluctuated frequently, which indicates that the factors affecting the concentration
variation were very complicated. The mean concentration of GEM during this period was 2.73 ng
m$^{-3}$, and the maximum difference of GEM diurnal variation was 1.40 ng m$^{-3}$. September was the
ending period of the Indian summer monsoon, and the external transboundary transport of air masses
by the monsoon may have been weakened. Simultaneously, the average temperature in September
was high (14.4°C) without residents heating during the night, which may be the reason for the small
diurnal variation in GEM concentrations. Overall, GEM concentrations remained lower during the
day, increased, and remained high at night. No clear pattern of variation was observed in GOM and
PBM during this period.
During the WEC1 period, the diurnal variation in GEM concentrations was exceedingly large.
The mean GEM value was 2.39 ng m$^{-3}$ with great GEM diurnal variation value (3.01 ng m$^{-3}$). The
diurnal trend was roughly the same as the whole monitoring period, but the lowest value was 1.06
ng m$^{-3}$ and the highest value was 4.07 ng m$^{-3}$. The increase in night concentrations may be related
to the beginning of the heating season, whereas the burning of yak dung, firewood, or other fuels
for residential heating may release GEM and PBM, and accumulate within the boundary layer at
night, leading to higher GEM concentrations (Rhode et al., 2007;Xiao et al., 2015;Chen et al., 2015).
In contrast, low day values may be related to the wind field. Although the wind speed during the
WEC1 period was similar to that during the S-ISM period, westerly circulation was prevalent in
Lhasa during the WEC1 period, with air masses coming from the sparsely populated area with lower
GEM concentrations. The westerly winds carry away the locally accumulated GEM air masses in
Lhasa and bring in low GEM concentration air masses from west of Lhasa. During this period, the
diurnal variation in GOM concentrations was not clear.
The influence of wind flow was evident during the WEC2 period. During this period, GEM
concentrations in Lhasa were the lowest and close to 1.00 ng m$^{-3}$ in the late afternoon when wind
speeds were the highest. Nighttime concentrations were lower in the WEC2 period than in the
WEC1 period, probably because wind speeds were also high at night during the WEC2 period
(19:00-4:00 UTC+8, mean wind speed 1.74 m/s). The heating season was still in progress during
the WEC2 period, but GEM emissions from heating were likely carried away by the continuous
flow of air masses. Therefore, the mean GEM concentration in Lhasa during the winter was likely
strongly influenced by wind speed.
Compared to the diurnal variation patterns obtained from monitoring in other remote areas of
China (Figure S3), the diurnal variation of GEM in Lhasa is more unique, and a wide diurnal
variation in concentration was observed. Valleys of GEM concentrations in the afternoon were also
observed in Waliguan and Namco, but the concentration variation ranges were small in both areas.
This could be related to the regional characteristics of the three sites. Waliguan is located in a
mountainous area with no significant local anthropogenic sources, and is mainly influenced by
valley winds (Fu et al., 2012a). Namco is located in the central Tibetan region away from
anthropogenic sources. Although the wind speed was also high in the afternoon in Namco (Yin et
al., 2018), it is likely that the diurnal variation in atmospheric GEM concentrations was small
because the atmosphere was well-proportioned and there were no local sources; therefore, the
changes in wind speed might not affect the GEM concentration. In contrast, with the difference in
GEM concentrations between Lhasa and the transmitted external air mass, wind speed could
significantly influence the GEM concentrations in Lhasa. Meteorological conditions may have
played a more important role. We conjecture that the high nighttime concentrations in Lhasa may
originate mainly from local emissions, while the high wind speed and mixing of clean external air
masses in the afternoon reduce the local GEM concentrations.
**3.4   Analysis of factors affecting atmospheric Hg concentration in Lhasa**
Overall, four components were resolved for each period, from S-ISM to WEC2. The Hg
monitoring data, meteorological factors, and pollutant data in Lhasa were statistically and
dimensionally reduced by PCA to analyze the relationships among them. The components related
to Hg were selected and analyzed separately for each period, and the principal components were
extracted (Table 4). According to the variable's loadings on each component, they were assigned as
Special Hg-related factor, Local emission factor, and Wind factor.





Table 4 PCA factor loadings (varimax rotated factor matrix) for Hg in Lhasa, China.

| Tentative Identification | Special Hg-related | | | Local emission | | Wind | |
|---|---|---|---|---|---|---|---|
| preiod | S-ISM | WEC1 | WEC2 | WEC1 | WEC2 | S-ISM | WEC2 |
| GEM | **0.87** | **0.63** | **0.9** | 0.44 | 0.3 | -0.21 | -0.13 |
| PBM | **0.96** | **0.79** | **0.63** | | | | |
| GOM | **0.95** | **0.87** | **0.95** | 0.11 | 0.1 | | |
| Temp | | | | -0.29 | 0.2 | | **0.92** |
| Hum | | -0.19 | 0.17 | -0.26 | -0.18 | -0.26 | **-0.75** |
| Wind_Speed | -0.1 | -0.14 | | -0.37 | | **0.87** | **0.91** |
| Rain | | | | | | | |
| Solar_Rad. | | | | -0.21 | | | |
| CO | 0.17 | | 0.11 | **0.87** | **0.69** | -0.22 | **0.51** |
| NO$_2$ | | 0.13 | | **0.92** | **0.96** | -0.25 | |
| O$_3$ | | | -0.36 | **-0.89** | **-0.83** | **0.53** | |
| PM$_{10}$ | -0.14 | | | **0.86** | **0.69** | | 0.14 |
| PM$_{2.5}$ | -0.16 | | 0.16 | **0.83** | **0.86** | | |
| SO$_2$ | 0.26 | 0.17 | **0.86** | **0.75** | | | |
| Variance Explained | 19.6 | 13.69 | 21.88 | 35.13 | 24.94 | 9.25 | 18.32 |

Note:variables with high factor loadings (> 0:5) were marked in bold. For readability, variables with very low factor loadings (< 0:1) are not presented.


A special Hg-related factor (Factor 1) was assigned owing to the continuous high positive
loading of GEM, GOM, and PBM from the SISM to the WEC periods; only high positive loading
of SO$_2$ was found in the WEC2 period. For this factor, there was no significant relationship with
any other meteorological factors or pollutants. The high positive loading of this factor for all three
Hg species and the low correlation with meteorological factors and other pollutants may indicate
that this is a specific source of Hg. As the short transmission distance of the GOM and PBM, the
special Hg-related source should be closer to Lhasa. However, no particular source of Hg around
Lhasa has been reported in the literature, so the source indicated by this factor remains unclear.
Factor 2 had slight positive relationship with GEM and GOM, high positive loading of
CO/NO$_2$/PM$_{10}$/PM$_{2.5}$/SO$_2$, and high negative loading of O$_3$ during the WEC period. The high
positive loadings of CO/NO$_2$/PM$_{10}$/PM$_{2.5}$/SO$_2$ may indicate that this factor is highly correlated with
the combustion source. As O$_3$ concentration rapidly decreased after sunset (Figure 4. a), high
negative loading of O$_3$ further indicated that this factor represented events during the nighttime with



low O₃ concentrations. Since the WEC period overlapped with the heating season in Lhasa, Factor

2 may be strongly associated with heating combustion.

The wind factor (Factor 3) involves high positive loading of wind speed and low negative

loading of GEM concentration during the S-ISM and WEC2 periods. This factor reveals the

scavenging effect of wind on the local GEM in Lhasa. For high wind speed, the air masses with low

GEM concentrations in the surrounding area mixed with the air masses in Lhasa, leding to a

reduction in the GEM concentration. Concurrently, some GEM in Lhasa was carried away from the

city by wind. Factor 3 was consistent with the analysis of the wind effect on the diurnal variation of

Hg concentrations (Section 3.2), indicating the effect of strong winds on urban Hg removal.

### 3.5 Atmospheric Hg source trajectories and potential source regions in the Lhasa area

Figure 4 shows the GEM backward trajectory paths from the S-ISM period to the WEC2 period

in Lhasa. During the S-ISM period, most trajectories (cluster 1, representative GEM concentration

of 2.66 ng m$^{-3}$, 87.53% of the trajectory during this period, Table S1) originated from or passed

through the south of Lhasa. The source of the trajectory points directly to the Indian Ocean, likely

as these transported air masses are still subject to Indian monsoon action in September. According

to the UNEP Hg emission inventory (UNEP, 2013), there are few anthropogenic emissions along

this trajectory, indicating that the GEM may originate from the Indian Ocean or locally from Lhasa.

Clusters 2 and 3 may indicate the trajectories of air masses driven by westerly circulation, which

had a low proportion in the S-ISM period, with slightly different GEM concentrations from different

sources.

During the WEC1 period, the driving factor of the air mass gradually shifted from Indian

monsoon to westerly circulation. Clusters 2 and 3 are trajectories driven by the higher-height

westerly circulation during the WEC1 period with a higher proportion than in the S-ISM period.

Cluster 1 came from the southwest of Lhasa, and the air mass moved along the Himalayas before

entering the Tibetan Plateau and was transported to Lhasa. Compared to the S-ISM period, the GEM

concentration of this trajectory decreased slightly, which may be related to the source of the air mass

and the areas it passes through. WEC2 showed little change in trajectory sources compared to WEC1,

but all trajectory concentrations decreased significantly. Both WEC1 and WEC2 were in winter,

and both had similar trajectories related to the driving wind field. Significantly decreasing GEM

concentrations may suggest a local influence in Lhasa City. The local GEM in Lhasa consists of
background concentrations superimposed on local emissions, and the share of local emissions
decreases under better dispersion conditions (higher wind speeds) during the WEC2 period. The
GEM concentration during the WEC2 period in Lhasa is only 0.16 ng m$^{-3}$, higher than the
concentration during the WEC period in the QNNP region (1:31 ± 0:42 ng m$^{-3}$) (Lin et al., 2019)).

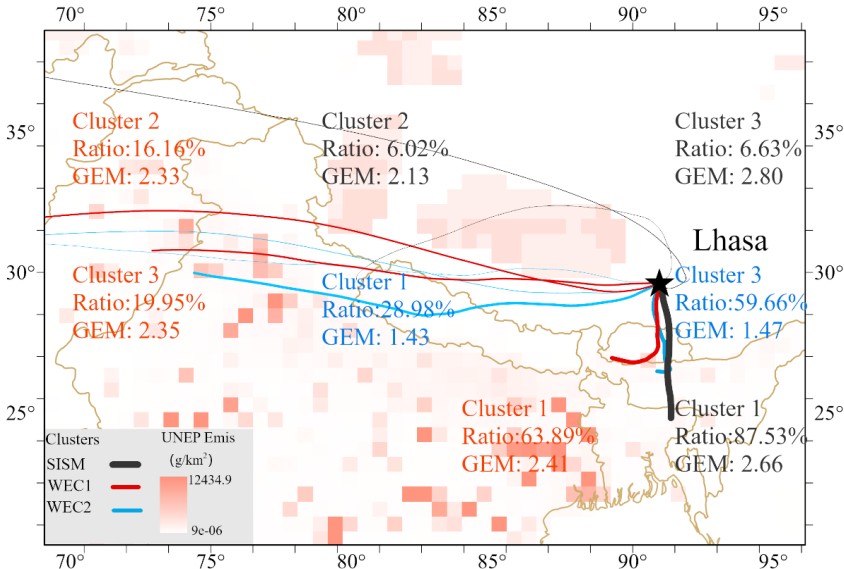

**Figure 5 Clusters of the back trajectory analysis from Lhasa during S-ISM and WEC**
**periods. The thickness of the line represents the ratio of the cluster in the time period, the**
**background is the global Hg emission inventory developed by UNEP (2013a).**
**3.6  Comparison of three typical sites on the Tibetan Plateau**
Different levels and variation patterns of Hg species concentrations were observed in the
Qomolangma National Nature Preserve (QNNP, 4,276 m) (Lin et al., 2019), Nyingchi (SET, 3,263
m) (Lin et al., 2022), and Lhasa (3,650 m) areas, which are typical monsoon-influenced, canyon,
and urban areas on the Tibetan Plateau, respectively. Predictably, Hg species concentrations were
generally lower on the Tibetan Plateau, while background areas with few populations on the Plateau
were lower than those on the plains in mainland China, and cities on the Plateau were lower than
those on the plains. Although the transboundary transport of pollutants has received considerable
notice (Huang et al., 2016b;Wang et al., 2018;Lin et al., 2019;Zhu et al., 2019;Feng et al., 2019),



the comparison between QNNP and Lhasa indicates that the contribution of local anthropogenic
sources may be significant for atmospheric pollutant concentrations. The comparison also suggests
that atmospheric Hg emissions from urban residential life may be an important source that can be
valuable in inventory studies.

483       Vastly different patterns of Hg species concentrations were found in the QNNP and SET due

to differences in geography and vegetation cover. Atmospheric Hg transported to QNNP is subject
to a combination of monsoonal action and the pumping effect of glacial winds. However,
atmospheric Hg entering the SET area has a slow elevation increasing path, and abundant
precipitation and vegetation have a significant trapping effect on atmospheric Hg. While for Lhasa,
a direct effect of local wind fields on the accumulation or rapid removal of GEM concentrations
was found. This suggests that although global transport is important in the Hg cycle, the pollution
at each location is likely to be strongly influenced by the local environment, even in exceptionally
clean areas such as the Tibetan Plateau. However, there are still relatively few studies on GOM and
PBM in the Tibetan Plateau, and more comprehensive investigations are required on how changes
in Hg speciation transformation on snow and ice surfaces affect the environment, the effect of
stratospheric intrusion on GOM concentrations (which is common on the Plateau), and the effect of
particulate matter on Hg(II) gas-particle partitioning, which would help to understand the Hg species
change on the Plateau and throughout the world.
**4. Conclusions**

498       Lhasa is the largest city on the Tibetan Plateau; thus, its atmospheric Hg concentrations

represent the highest level of atmospheric Hg pollution in this area. Unexpectedly high
concentrations of atmospheric mercury species were found in Lhasa. The GEM concentrations were
higher than the Northern Hemisphere background concentrations, and the GOM and PBM
concentrations were high among Chinese cities. Monitoring of atmospheric Hg in Lhasa showed
that the mean concentrations of GEM, GOM, and PBM during the S-ISM period ($2.73 \pm 1.48$ ng m$^-$
$^3$, $38.4 \pm 62.7$ pg m$^{-3}$, and $59.1 \pm 181.0$ pg m$^{-3}$, respectively) were higher than those during the WEC
period ($2.11 \pm 2.09$ ng m$^{-3}$, $35.8 \pm 43.3$ pg m$^{-3}$, and $52.9 \pm 90.1$ pg m$^{-3}$, respectively). Combined
with the trajectory analysis, the high atmospheric Hg concentrations during the S-ISM phase may
have originated from external long-range transport.


Analysis of the overall concentration changes revealed some irregular and sudden high
atmospheric Hg concentration events in Lhasa. Analysis of these events suggests that local sources
(such as combustion events) can cause severely elevated concentration events under low wind
speeds and the presence of a low-height nighttime boundary layer. Analysis of the diurnal variation
of concentrations confirmed that low wind speeds and a low height nocturnal boundary layer would
lead to the elevated local Hg concentrations. In contrast, higher wind speeds could rapidly remove
atmospheric Hg from Lhasa. PCA analysis of the influencing factors indicates that local sources,
especially special Hg-related sources, are important factors influencing the variability of
atmospheric Hg. The PCA analysis also indicated the important role of higher wind speeds in
reducing atmospheric Hg concentrations in the urban areas of Lhasa, likely owing to the large Hg
concentration difference between Lhasa and surrounding areas.
Up to this study, we have obtained atmospheric Hg monitoring data from four typical areas of
the Tibetan Plateau: Lhasa, QNNP, SET, and Namco. The atmospheric Hg concentrations in the
background areas were at or below the average GEM concentration of Northern Hemisphere, with
higher levels in the urban area of Lhasa. Factors such as long-range transport of atmospheric
mercury, effects of local meteorological conditions, local glaciers, etc., were considered in these
studies. Further monitoring of additional areas and regional simulations are required to confirm the
atmospheric Hg transport patterns and fluxes.

**Acknowledgments**
This study was funded by the National Natural Science Foundation of China (Grant #41821005,
41977311, 42122059). The authors are grateful to NOAA for providing the HYSPLIT model and
GFS meteorological files. We also thank the staffs of Lhasa station of the Institute of Tibetan Plateau
Research, Chinese Academy of Sciences on Lhasa for field sampling assistance.
**Data availability.** All the data presented in this paper can be made available for scientific
purposes upon request to the corresponding authors.
**Author contributions.** HL,XW, YT, QZ and XY designed the research and performed field
measurements. HL and YT performed the data analysis and model simulations. HL led the paper
writing. LC,CY, ZC, QZ, SK, JL, JS and BF contributed to the scientific discussion and the paper



preparation.

538        **Competing interests.** The authors declare that they have no conflict of interest.






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
