# Peer review of "Unexpectedly high concentrations of atmospheric mercury"

_Atmospheric Chemistry and Physics, 2022_

## Author Comment (AC1)

Responses to the Reviewers' Comments

**Unexpectedly high concentrations of atmospheric mercury species in Lhasa, the largest city on the Tibetan Plateau**

Dear editor and reviewer,

We greatly appreciate the useful comments and suggestions from the editor and reviewers. We think the novelty and importance of this study have been acknowledged by the reviewers. We have revised the manuscript thoroughly based on the reviewers' comments. Detailed point by point responses are provided below. All the revisions have been highlighted in blue in the revised manuscript. We hope the revised manuscript could meet the standard of ACP. Thanks again for your consideration.

**Referee #1**

**General comments:**

This paper presents continuous observations of different atmospheric mercury species in Lhasa, the largest city on the Tibetan plateau. Atmospheric mercury was analyzed for different periods (Indian Summer Monsoon and Westerly Circulation) and further studied the daily variation of atmospheric mercury concentration as well as some typical high-value cases. The results indicate that the atmospheric mercury concentration in Lhasa is significantly higher than that in the adjacent region, reflecting the influence of local anthropogenic emissions and wind fields on the atmospheric mercury concentration in remote areas. Overall, this manuscript is clear written and makes up for the lack of observations of atmospheric mercury at high altitudes. The paper can be accepted after addressing the following comments.

**Response:**

Thanks for your detailed comments and suggestions. We have reviewed the paper in detail according to the reviewer's comments and made corresponding modifications accordingly. Please see the revised manuscript. All the revisions have been highlighted in blue. Detailed responses to your comments are provided as follows.

Specific comments:

**Comment #1**

Lines 91-92, "whether the Tibetan Plateau can be treated as a background area for studying atmospheric Hg". It's an interesting question, and what do you think about it after you've made several observations of atmospheric Hg (such as Lhasa, QNNP, SET, and Namco) on the Tibetan Plateau?

**Response #1**

Thanks for the suggestion. Based on the available atmospheric Hg monitoring data at QNNP, Nyingchi, and Lhasa, we believe that, in general, Tibetan Plateau can be regarded as a background area for atmospheric Hg study, except for urban areas.

**Comment #2**

Line 100, please add a reference for the average GEM concentrations in the Northern Hemisphere.

**Response #2**

Thanks for the suggestion. We have added '**(Lindberg et al., 2007; Slemr et al., 2015; Venter et al., 2015; Sprovieri et al., 2016; Lan et al., 2012)**' as references for the average GEM concentrations in the Northern Hemisphere.

**Comment #3**

Lines 110-111, "the wet deposition of total Hg and particulate Hg was higher during the non-monsoon period than during the monsoon period". The wet deposition is correlated with the precipitation, but why is Hg wet deposition lower during the monsoon than during the non-monsoon?

**Response #3**

Thanks for the comment. The comparison here is about Hg concentration rather than Hg deposition flux. The sentence has been revised as: '**Huang et al. (2013) measured the wet deposition of atmospheric Hg in Lhasa in 2010 and showed that the wet depositions of total Hg concentration and particulate Hg concentration were higher during the non-monsoon period than that during the monsoon period.**'

**Comment #4**

L116, "continuous"

**Response #4**

We have replaced the word accordingly. Thanks for the suggestion.

**Comment #5**

L195, "categorized"

**Response #5**

We have replaced the word accordingly. Thanks for the suggestion.

**Comment #6**

Fig.1, please add mark (°N, °E) to the latitude and longitude of the coordinate axis. Same for Fig. 5.

**Response #6**

Thanks for the suggestion. We have added marks of latitude and longitude in Fig.1 and Fig. 5.

**Comment #7**

Lines 217-219, GEM concentrations are nearly twice as high in September as in February, contrary to previous studies (e.g., Horowitz et al. 2017; Jiskra et al., 2018) that atmospheric mercury in the Northern Hemisphere is low in summer and higher in winter. Please try to explain the reason.

**Response #7**

Thanks for the suggestion. GEM concentrations showed a decreasing trend throughout the monitoring period, with the average weekly concentration decreasing from 3.21 ng m$^{-3}$ at the beginning of the monitoring period to 1.60 ng m$^{-3}$ at the end of the monitoring period. The different variation trends between Lhasa and the whole Northern Hemisphere may be related to the special location of Lhasa. GEM concentrations in Lhasa in September and in February may be related to different Hg source regions and Hg sources. In September, GEM in Lhasa may be influenced by Indian monsoon and receive transboundary transported GEM. While in February, GEM in Lhasa are potentially related to local anthropogenic emissions. We have added here a comparison with the trend of atmospheric GEM concentrations in the Northern Hemisphere. The added text is: '**Overall, GEM concentrations showed a decreasing trend throughout the monitoring period, with the average weekly concentrations decreasing from 3.21 ng m$^{-3}$ at the beginning of the monitoring period to 1.60 ng m$^{-3}$ at the end of the monitoring period, which is contrast to previous studies, which showed that atmospheric Hg concentrations in the Northern Hemisphere is low in summer and high in winter (Horowitz et al., 2017; Jiskra et al., 2018). The different variation trends between Lhasa and the whole Northern Hemisphere may be related to the special location of Lhasa.**'

**Comment #8**

Line 237, please give the definition of WEC1 and WEC2 in the paper.

**Response #8**

Thanks for the suggestion. We have added the definitions of S-ISM, WEC, WEC1, and WEC2 in Section 2.1. The sentence is revised as: '**The monitoring in Lhasa included the subsequent Indian Summer Monsoon (S-ISM, August 29 to September 30, 2016) and Westerly Circulation (WEC, October 1, 2016 to February 2, 2017) periods from August 29, 2016 to February 2, 2017. To better understand the changes of atmospheric Hg concentrations in different periods, the WEC period was further divided into WEC1 (October 1 to December 30, 2016) and WEC2 (January 1 to February 2, 2017) periods.**'

**Comment #9**

Lines 242-243, "the GEM concentration in Lhasa is low among the provincial capitals in China." is inconsistent with "The atmospheric Hg species concentrations were higher than ... other provincial capitals in China." in Lines 33-34.

**Response #9**

Thanks for the comment. The sentence in Lines 33-34 has been revised as: '**The GOM and PBM concentrations were higher than those of previous monitoring on the Tibetan Plateau and other provincial capitals in China.**'

**Comment #10**

2, What is the reason for the sparse observations in 2016/12?

**Response #10**

Thanks for the comment. The sparse observation in 2016/12 was caused by the Tekran instrument failure.

**Comment #11**

3, please add the meaning of the dotted lines and the units of different observations.

**Response #11**

Thanks for the suggestions. We have added units of different observations in Figure 3. The dotted lines indicate the range where the concentrations of Hg are significantly elevated. We have added an explanation of dotted line in the title of the figure, as follows: '**Figure 3 Comparison of the pollutant concentrations with those at the two days before and after the high Hg concentration events. The dotted lines indicate the area where the Hg concentrations are significantly elevated.**'

**Comment #12**

Section 3.2, it will be more intuitive to add a correlation analysis between Hg and other pollutant species? In addition, ozone, aerosols, and NO2 are thought to be related to Hg chemistry, and their relationship in plateau would be meaningful to the study of Hg.

**Response #12**

Thanks for the suggestions.

In Section 3.2, we did not use correlation analysis primarily because the time ranges over the atmospheric Hg concentration elevation events are short. The data for other pollutant species corresponding to the events are insufficient for correlation testing. The analysis of atmospheric Hg concentration elevation events in this section is intended to identify possible sources of atmospheric Hg through changes of concentrations of other pollutants over the same time period, and the changes of concentrations of other pollutants collected during the events are not precise enough to carry out an analysis of Hg chemistry. In future studies, we may make more precise measurements of the concentrations of other pollutants simultaneously and explore their relationship to Hg chemistry.

**Comment #13**

4a, please add units of the pollutants.

**Response #13**

Thanks for the suggestion. We have added units of different observations in Figure 4a.

**Comment #14**

Line 413, what are the "four components"?

**Response #14**

Thanks for the question. The components here refer to the factors obtained from PCA analyze for the periods from S-ISM to WEC2. We have rewritten this sentence to make them clearer. The

revised sentence is: '**Overall, four principal components were obtained for each period, from S-ISM to WEC2, using PCA analyze, to analyze the relationships between Hg and multiple pollutants and meteorological variables.**'

**Comment #15**

Table 4 note, please correct "> 0:5" & "<0:1".

**Response #15**

We have replaced the punctuation accordingly. Thanks for the suggestions.

**Comment #16**

Lines 465-466, "The GEM concentration WEC2 in Lhasa is 0.16 ng m$^{-3}$, higher than 1.31 in QNNP?" And the "1:31 ± 0:42 ng m$^{-3}$". Please correct.

**Response #2**

Thanks for the suggestions. We have rewritten the sentence: '**The GEM concentration during the WEC2 period in Lhasa was only 0.16 ng m$^{-3}$ higher than that during the WEC period in the QNNP region (1.31 ± 0.42 ng m$^{-3}$) (Lin et al., 2019)).**'

**References**

Horowitz, H. M., Jacob, D. J., Zhang, Y., Dibble, T. S., Slemr, F., Amos, H. M., Schmidt, J. A., Corbitt, E. S., Marais, E. A., and Sunderland, E. M.: A new mechanism for atmospheric mercury redox chemistry: Implications for the global mercury budget, Atmospheric Chemistry and Physics, 17, 6353-6371, 2017.

Jiskra, M., Sonke, J. E., Obrist, D., Bieser, J., Ebinghaus, R., Myhre, C. L., Pfaffhuber, K. A., Wängberg, I., Kyllönen, K., and Worthy, D.: A vegetation control on seasonal variations in global atmospheric mercury concentrations, Nature Geoscience, 11, 244-250, 2018.

Lan, X., Talbot, R., Castro, M., Perry, K., and Luke, W.: Seasonal and diurnal variations of atmospheric mercury across the US determined from AMNet monitoring data, Atmospheric Chemistry and Physics, 12, 10569-10582, 2012.

Lin, H., Tong, Y., Yin, X., Zhang, Q., Zhang, H., Zhang, H., Chen, L., Kang, S., Zhang, W., and Schauer, J.: First measurement of atmospheric mercury species in Qomolangma Natural Nature Preserve, Tibetan Plateau, and evidence of transboundary pollutant invasion, Atmospheric Chemistry and Physics, 19, 1373-1391, 2019.

Lindberg, S., Bullock, R., Ebinghaus, R., Engstrom, D., Feng, X., Fitzgerald, W., Pirrone, N., Prestbo, E., and Seigneur, C.: A synthesis of progress and uncertainties in attributing the sources of mercury in deposition, AMBIO: a Journal of the Human Environment, 36, 19-33, 2007.

Slemr, F., Angot, H., Dommergue, A., Magand, O., Barret, M., Weigelt, A., Ebinghaus, R., Brunke, E.-G., Pfaffhuber, K. A., and Edwards, G.: Comparison of mercury concentrations measured at several sites in the Southern Hemisphere, Atmospheric Chemistry and Physics, 15, 3125-3133, 2015.

Sprovieri, F., Pirrone, N., Bencardino, M., D'Amore, F., Carbone, F., Cinnirella, S., Mannarino, V., Landis, M., Ebinghaus, R., and Weigelt, A.: Atmospheric mercury concentrations observed at ground-based monitoring sites globally distributed in the framework of the GMOS network, Atmospheric Chemistry and Physics, 16, 11915-11935, 2016.

Venter, A., Beukes, J., Van Zyl, P., Brunke, E.-G., Labuschagne, C., Slemr, F., Ebinghaus, R., and Kock, H.: Statistical exploration of gaseous elemental mercury (GEM) measured at Cape Point from 2007 to 2011, Atmospheric Chemistry and Physics, 15, 10271-10280, 2015.

---

## Author Comment (AC2)

**Responses to the Reviewers' Comments**
**Unexpectedly high concentrations of atmospheric mercury species in Lhasa, the largest city on the Tibetan Plateau**

Dear editor and reviewer,

We greatly appreciate the useful comments and suggestions from the editor and reviewers. We think the novelty and importance of this study have been acknowledged by the reviewers. We have revised the manuscript thoroughly based on the reviewers' comments. Detailed point by point responses are provided below. All the revisions have been highlighted in blue in the revised manuscript. We hope the revised manuscript could meet the standard of ACP. Thanks again for your consideration.

**Anonymous Referee #2**

This article entitled 'Unexpectedly high concentrations of atmospheric mercury species in Lhasa, the largest city on the Tibetan Plateau' by Lin et al. analyzes the monitoring concentrations and source analysis of atmospheric mercury in important cities on the Tibetan Plateau. According to the local meteorological conditions, the author divided the monitoring period into two periods. Indian Summer Monsoon and Westerly Circulation. The authors analyzed the unusual phases of high atmospheric mercury concentration events and attempted to analyze the causes, and also analyzing the diurnal variation of atmospheric mercury in Lhasa. This article reveals the changes of urban atmospheric mercury in clean background areas, indicating the impact of urban anthropogenic activities and wind speed on regional atmospheric mercury concentrations. The result makes sense, and the output of the study will be useful for future studies on global mercury cycling and modeling. The manuscript is generally well organized and written. After revising the questions listed below, the study is accepted for publication.

**Response:**

Thanks for your detailed comments and suggestions. We have reviewed the paper in detail according to the reviewer's comments and made corresponding modifications accordingly. Please see the revised manuscript. All the revisions have been highlighted in blue. Detailed responses to your comments are provided as follows.

**Specific comments**

**Comment #1**

In line 175, the author only calculated the backward trajectory of GEM in the article, why did the author not consider analyzing the trajectory of GOM and PBM.

**Response #1**

Thanks for the comment. In this manuscript, we carried out trajectory analysis for GEM in Lhasa. Considering the complex topography of the Tibetan Plateau and the fact that most of the trajectories would pass through dramatic elevation rise when entering Tibetan Plateau, where the subsidence of GOM/PBM is very complex, we think that backward trajectory simulations of GOM and PBM in

Lhasa may introduce considerable errors and uncertainties.

**Comment #2**

Line 226, the author should show the timing of ISM and WEC in Figure 2.

**Response #2**

Thanks for the suggestion. We have added time markers to Figure 2.

**Comment #3**

Line 240, Table 1, what is the purpose of dividing WEC into WEC1 and WEC2?

**Response #3**

Thanks for the question. To better understand the changes of atmospheric Hg concentrations in different periods, the WEC period was further divided into WEC1 and WEC2 periods. We have added some explanations in Section 2.1. '**To better understand the changes of atmospheric Hg concentrations in different periods, the WEC period was further divided into WEC1 (October 1 to December 30, 2016) and WEC2 (January 1 to February 2, 2017) periods.**'

**Comment #4**

Line 229-231, is it possible that the change in the relationship between the concentration of GOM and PBM is related to the change in the gas-solid distribution ratio of atmospheric mercury?

**Response #4**

Thanks for the comment. We agree with the reviewers that temperature, humidity, and particulate matter composition may affect the partitioning of Hg(II) between GOM and PBM. During the period transiting from ISM to WEC, the climate and atmospheric composition of Lhasa changed significantly, which is likely to affect the partition ratio of GOM and PBM. We have added some relevant discussions in the paper. '**GOM and PBM may undergo mutual transformation in the atmosphere, which may be related to temperature, humidity, and atmospheric composition (Rutter and Schauer, 2007; Rutter et al., 2008), and thus the concentration distributions of GOM and PBM may also be related to the changes of local climate and atmospheric composition from S-ISM to WEC periods.**'

**Comment #5**

Line 268& 440, the author claims that strong winds may reduce the concentration of atmospheric mercury in the city. I wonder how the atmospheric mercury concentration in Lhasa compares with the background area (such as Nam Co) under the condition of strong winds?

**Response #5**

Thanks for the comment. We counted the GEM concentration data at the period with wind speed greater than 4 m s$^{-1}$, and the GEM concentration was $1.31 \pm 0.93$ ng m$^{-3}$, similar to that of Nam Co at $1.33 \pm 0.24$ ng m$^{-3}$. We have added the relevant comparison in Section3.4. '**At wind speed greater than 4m s$^{-1}$, the average GEM concentration in Lhasa was 1.31±0.93 ng m$^{-3}$, which is similar**

**to the average concentration of Nam Co (1.33±0.24 ng m$^{-3}$) in the Tibetan hinterland.'**

**Comment #6**

Line 297, how did the authors confirm that the high mercury concentration events occurred randomly?

**Response #6**

Thanks for the question. The observed high Hg concentration events occurred at different time intervals, in different forms of concentration change curves, and the values of high Hg concentrations were also different, so we concluded that the high Hg concentration events occurred randomly.

**Comment #7**

Line 323 Table 3, Please note that colored values in tables are not allowed in ACP, please consider replacing them with italic or bold lettering.

**Response #7**

Thanks for the suggestions. Revisions have been made accordingly.

**Comment #8**

Line 396, does the author think that the changes in atmospheric mercury in Lhasa can be extended to more cities?

**Response #8**

Thanks for the suggestions. Based on the analysis in this paper, we believe that the effects of anthropogenic source emissions of atmospheric Hg on urban atmospheric Hg concentrations are widespread and may occur in other cities as well. However, the ability of wind fields to remove urban Hg pollutants depends on the strength and source of the wind, and only clean wind sources and sufficiently strong and persistent wind fields can remove urban Hg pollution. This hypothesis needs to be confirmed by monitoring atmospheric Hg concentrations in other cities and surrounding areas.

References

Rutter, A. P. and Schauer, J. J.: The impact of aerosol composition on the particle to gas partitioning of reactive mercury, Environmental science & technology, 41, 3934-3939, 2007.

Rutter, A. P., Schauer, J. J., Lough, G. C., Snyder, D. C., Kolb, C. J., Von Klooster, S., Rudolf, T., Manolopoulos, H., and Olson, M. L.: A comparison of speciated atmospheric mercury at an urban center and an upwind rural location, Journal of Environmental Monitoring, 10, 102-108, 2008.